# SpectroMotion: Dynamic 3D Reconstruction of Specular Scenes

## Abstract

We present SpectroMotion, a novel approach that combines 3D Gaussian Splatting (3DGS) with physically-based rendering (PBR) and deformation fields to reconstruct dynamic specular scenes. Previous methods extending 3DGS to model dynamic scenes have struggled to accurately represent specular surfaces. Our method addresses this limitation by introducing a residual correction technique for accurate surface normal computation during deformation, complemented by a deformable environment map that adapts to time-varying lighting conditions. We implement a coarse-to-fine training strategy that significantly enhances both scene geometry and specular color prediction. We demonstrate that our model outperforms prior methods for view synthesis of scenes containing dynamic specular objects and that it is the only existing 3DGS method capable of synthesizing photorealistic real-world dynamic specular scenes, outperforming state-of-the-art methods in rendering complex, dynamic, and specular scenes.

## 1 Introduction

3D Gaussian Splatting (3DGS) (Kerbl et al., 2023) has emerged as a groundbreaking technique in 3D scene reconstruction, offering fast training and real-time rendering capabilities. By representing 3D space using a collection of 3D Gaussians and employing a point-based rendering approach, 3DGS has significantly improved efficiency in novel-view synthesis. However, extending 3DGS to accurately model dynamic scenes, especially those containing specular surfaces, has remained a significant challenge.

Existing extensions of 3DGS have made progress in either dynamic scene reconstruction or specular object rendering, but none have successfully combined both aspects. Methods tackling dynamic scenes often struggle with accurate representation of specular surfaces, while those focusing on specular rendering are limited to static scenes. This gap in capabilities has hindered the application of 3DGS to real-world scenarios where both motion and specular reflections are present.

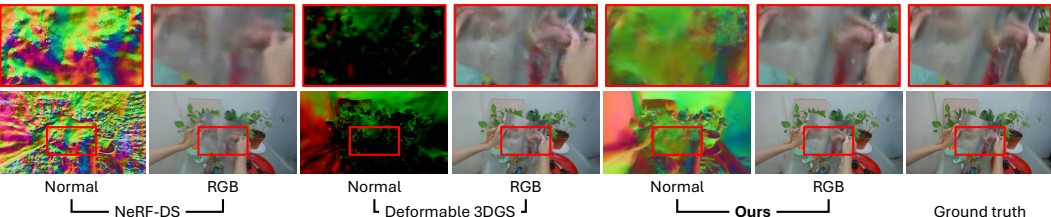

Figure 1: **Our method, SpectroMotion, recovers and renders dynamic scenes with higher-quality reflections compared to prior work.** It introduces physical normal estimation, deformable environment maps, and a coarse-to-fine training strategy to achieve superior results in rendering dynamic scenes with reflections. Here we present a rendered test image along with its corresponding normal maps and a ground-truth image. For Deformable 3DGS, we use the shortest axes of the deformed 3D Gaussians as the normals. We have highlighted the specular regions for a scene from the NeRF-DS dataset (Yan et al., 2023) to demonstrate the effectiveness of our approach.

We present SpectroMotion, a novel approach that addresses these limitations by combining 3D Gaussian Splatting with physically-based rendering (PBR) and deformation fields. Our method introduces three key innovations: a residual correction technique for accurate surface normal computation during deformation, a deformable environment map that adapts to time-varying lighting conditions, and a coarse-to-fine training strategy that significantly enhances both scene geometry and specular color prediction.

Our evaluations demonstrate that SpectroMotion outperforms prior methods in view synthesis of scenes containing dynamic specular objects, as illustrated in Figure 1. It is the only existing 3DGS method capable of synthesizing photorealistic real-world dynamic specular scenes, surpassing state-of-the-art techniques in rendering complex, dynamic, and specular content. This advancement represents a significant leap in 3D scene reconstruction, particularly for challenging scenarios involving moving specular objects.

In summary, we make the following contributions:

- We propose SpectroMotion, a physically-based rendering (PBR) approach combining deformation fields and 3D Gaussian Splatting for real-world dynamic specular scenes.

- We introduce a residual correction method for accurate surface normals during deformation, coupled with a deformable environment map to handle time-varying lighting conditions in dynamic scenes.

- We develop a coarse-to-fine training strategy enhancing scene geometry and specular color prediction, outperforming state-of-the-art methods.

## 2 RELATED WORK

**Dynamic Scene Reconstruction.** Recent works have leveraged NeRF representations to jointly solve for canonical space and deformation fields in dynamic scenes using RGB supervision (Guo et al., 2023; Li et al., 2021; Park et al., 2021a;b; Pumarola et al., 2020; Tretschk et al., 2021; Xian et al., 2021). Further advancements in dynamic neural rendering include object segmentation (Song et al., 2023), incorporation of depth information (Attal et al., 2021), utilization of 2D CNNs for scene priors (Lin et al., 2022; Peng et al., 2023), and multi-view video compression (Li et al., 2022). However, these NeRF-based methods are computationally intensive. To address this, recent research has adapted 3D Gaussians for dynamic scenes (Yang et al., 2023c; Wu et al., 2023; Huang et al., 2024; Liang et al., 2023c; Wang et al., 2024; Mihajlovic et al., 2024; Stearns et al., 2024), primarily focusing on deforming spatial coordinates through deformation fields. Nevertheless, these approaches do not explicitly account for changes in object surface during the deformation process. Our work extends this line of research by combining specular object rendering based on normal estimation with a deformation field, enabling each 3D Gaussian to effectively model dynamic specular scenes.

**Reflective Object Rendering.** While significant progress has been made in rendering reflective objects, challenges arising from complex light interactions persist. Recent years have seen numerous studies addressing these issues, primarily by decomposing appearance into lighting and material properties (Bi et al., 2020; Boss et al., 2021; Li & Li, 2022; Srinivasan et al., 2020; Zhang et al., 2021b; Munkberg et al., 2022; Zhang et al., 2021a; Verbin et al., 2024a; Zhao et al., 2024). Building on this foundation, some research has focused on improving the capture and reproduction of specular reflections (Verbin et al., 2022; Ma et al., 2023; Verbin et al., 2024b), while others have leveraged signed distance functions (SDFs) to enhance normal estimation (Ge et al., 2023; Liang et al., 2023a;b; Liu et al., 2023; Zhang et al., 2023). The emergence of 3D Gaussian Splatting (3DGS) has sparked a new wave of techniques (Jiang et al., 2023; Liang et al., 2023d; Yang et al., 2024; Ye et al., 2024; Zhu et al., 2024; Shi et al., 2023) that integrate Gaussian splatting with physically-based rendering. Nevertheless, accurately modeling dynamic environments and time-varying specular reflections remains a significant challenge. To address this limitation, our work introduces a novel approach incorporating a deformable environment map and additional explicit Gaussian attributes, specifically designed to capture specular color changes over time.

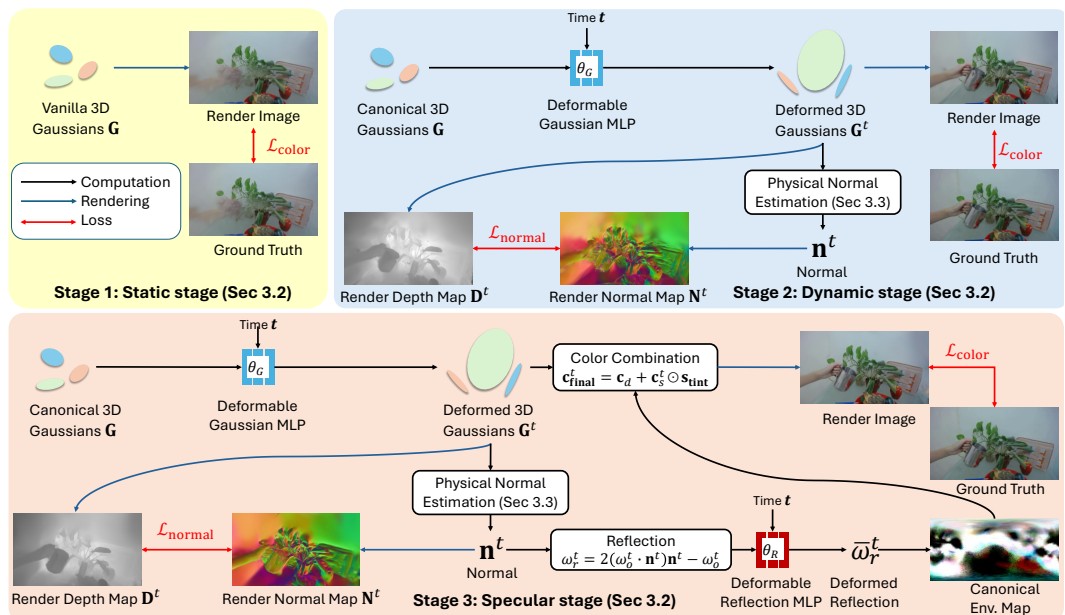

Figure 2: **Method Overview.** Our method stabilizes the scene geometry through three stages. In the static stage, we stabilize the geometry of the static scene by minimizing photometric loss $\mathcal{L}_{\text{color}}$ between vanilla 3DGS renders and ground truth images. The dynamic stage combines canonical 3D Gaussians $\mathbf{G}$ with a deformable Gaussian MLP to model dynamic scenes while simultaneously minimizing normal loss $\mathcal{L}_{\text{normal}}$ between rendered normal map $\mathbf{N}^t$ and gradient normal map from depth map $\mathbf{D}^t$, thus further enhancing the overall scene geometry. Finally, the specular stage introduces a deformable reflection MLP to handle changing environment lighting, deforming reflection directions $\omega_r^t$ to query a canonical environment map for specular color $\mathbf{c}_s^t$. It is then combined with diffuse color $\mathbf{c_d}$ (using zero-order spherical harmonics) and learnable specular tint $\mathbf{s_{tint}}$ per 3D Gaussian to obtain the final color $\mathbf{c}_{\text{final}}^t$. This approach enables the modeling of dynamic specular scenes and high-quality novel view rendering.

## 3 METHOD

**Overview of the approach.** The overview of our method is illustrated in Fig. 2. Given an input monocular video sequence of frames and corresponding camera poses, we design a three-stage approach to reconstruct the dynamic specular scene, as detailed in Section 3.2. Accurate specular reflection requires precise normal estimation, so Section 3.3 elaborates on how we estimate normals in dynamic scenes. Finally, we introduce the losses used throughout the training process in Section 3.4.

### 3.1 PRELIMINARY

**3D Gaussian Splatting.** Each 3D Gaussian is defined by a center position $\boldsymbol{x} \in \mathbb{R}^3$ and a covariance matrix $\boldsymbol{\Sigma}$. 3D Gaussian Splatting (Kerbl et al., 2023) optimizes the covariance matrix using scaling factors $\boldsymbol{s} \in \mathbb{R}^3$ and rotation unit quaternion $\boldsymbol{r} \in \mathbb{R}^4$. For novel-view rendering, 3D Gaussians are projected onto 2D camera planes using differentiable splatting (Yifan et al., 2019):

$$\boldsymbol{\Sigma}' = \mathbf{J}\mathbf{W}\boldsymbol{\Sigma}\mathbf{W}^T\mathbf{J}^T. \tag{1}$$

Pixel colors are computed using point-based volumetric rendering:

$$C = \sum_{i \in N} T_i \alpha_i c_i, \quad \alpha_i = \sigma_i e^{-\frac{1}{2}(\boldsymbol{x})^T \boldsymbol{\Sigma}'(\boldsymbol{x})}, \tag{2}$$

where $T_i = \prod_{j=1}^{i-1}(1 - \alpha_j)$ is the transmittance, $\sigma_i$ is the opacity, and $\mathbf{c}_i$ is the color of each 3D Gaussian.

## 3.2 SPECULAR RENDERING

**Limitations of existing methods.** The current Dynamic 3DGS-based methods (Wu et al., 2023; Yang et al., 2023c;b) encounter limitations in accurately modeling environments that include specular objects. This issue arises from the inherent low-frequency characteristics of low-order spherical harmonics (SH), which are inadequate for capturing complex visual effects such as specular highlights. In contrast, other specialized 3DGS-based methods for static specular object scenes (Jiang et al., 2023; Liang et al., 2023d) often incorporate environment maps to model lighting, which is then combined with BRDF to simulate the entire scene. However, vanilla environment maps are not suitable for modeling lighting scenarios that involve time-variant elements. This results in the existing 3DGS-based methods being insufficient for effectively modeling dynamic specular object scenes.

**Proposed solution overview.** To address these challenges, we introduce physical normal estimation (Section 3.3) and deformable environment maps to model the specular color of real-world dynamic scenes. However, this approach alone is insufficient, as precise scene geometry is crucial for accurate reflections. Therefore, we introduce our coarse-to-fine training strategy, which helps stabilize scene geometry while simultaneously predicting accurate specular color. Our coarse-to-fine training strategy is divided into three stages: the static stage, the dynamic stage, and the specular stage. In the following paragraphs, we will introduce each of these stages in detail.

### 3.2.1 COARSE-TO-FINE TRAINING STRATEGY

**Static stage.** In the static stage, we employ vanilla 3DGS (Kerbl et al., 2023) for static scene reconstruction to stabilize the geometry of the static scene. Specifically, we optimize the position $x$, scaling $s$, rotation $r$, opacity $\alpha$, and coefficients of spherical harmonics (SH) of the 3D Gaussians by minimizing the photometric loss $\mathcal{L}_{\text{color}}$ between the rendered image and the corresponding image:

$$\mathcal{L}_{\text{color}} = (1 - \lambda_{\text{D-SSIM}})\mathcal{L}_1 + \lambda_{\text{D-SSIM}}\mathcal{L}_{\text{D-SSIM}}. \tag{3}$$

**Dynamic stage.** Following the static stage, we address dynamic objects using Deformable 3DGS (Yang et al., 2023c). For each 3D Gaussian in canonical 3D Gaussians $\mathbf{G}$, we input its position $x$ and time $t$ into a deformable Gaussian MLP with parameters $\theta_G$ to predict position, rotation, and scaling residuals:

$$(\Delta\boldsymbol{x}^t, \Delta\boldsymbol{r}^t, \Delta\boldsymbol{s}^t) = F_{\theta_G}(\gamma(x), \gamma(t)), \tag{4}$$

where $\gamma$ denotes positional encoding. Attributes of the corresponding 3D Gaussian in deformed 3D Gaussians $\mathbf{G}^t$ at time $t$ is obtained by:

$$(\boldsymbol{x}^t, \boldsymbol{r}^t, \boldsymbol{s}^t) = (\Delta\boldsymbol{x}^t, \Delta\boldsymbol{r}^t, \Delta\boldsymbol{s}^t) + (\boldsymbol{x}, \boldsymbol{r}, \boldsymbol{s}). \tag{5}$$

This approach separates motion and geometric structural learning, allowing accurate simulation of dynamic behaviors while maintaining a stable geometric reference. To further enhance scene geometry, we estimate normals of deformed 3D Gaussians and optimize them using:

$$\mathcal{L}_{\text{normal}} = 1 - \mathbf{N}^t \cdot \hat{\mathbf{N}}^t, \tag{6}$$

where $\mathbf{N}^t$ is the rendered normal map and $\hat{\mathbf{N}}^t$ is the normal map derived from the rendered depth map $\mathbf{D}^t$. This process improves local associations among 3D Gaussians and optimizes both depth and normal information across the entire scene.

**Specular stage.** We adopt an image-based lighting (IBL) model with a learnable cube map. Inspired by the rendering equation (Kajiya, 1986), split-sum approximation (Karis & Games, 2013), and Cook-Torrance reflectance model (Cook & Torrance, 1982), we formulate the outgoing radiance of the specular component $L_s$ as:

$$L_s = \int_\Omega \frac{DGF}{4(\omega_o^t \cdot \mathbf{n}^t)(\omega_i \cdot \mathbf{n}^t)}(\omega_i \cdot \mathbf{n}^t)d\omega_i \int_\Omega L_i(\omega_i)D(\omega_i, \omega_o^t)(\omega_i \cdot \mathbf{n}^t)d\omega_i, \tag{7}$$

where $\Omega$ is the hemisphere around the surface normal $\mathbf{n}^t$. $D$, $G$, and $F$ represent the GGX normal distribution function (Walter et al., 2007), geometric attenuation, and Fresnel term, respectively. $\omega_o^t$ is the view direction, and $L_i(\omega_i)$ is the incident radiance. The first term, representing the specular BSDF with a solid white environment light, is precomputed and stored in a look-up table. The second term is

pre-integrated in a filtered cubemap, where each mip-level corresponds to a specific roughness value. Roughness $\rho \in [0, 1]$ is a learnable parameter for each 3D Gaussian. After the static and dynamic stages, the geometry is well-defined. This allows us to accurately calculate reflection directions $\omega_r^t$:

$$\omega_r^t = 2(\omega_o^t \cdot \mathbf{n}^t)\mathbf{n}^t - \omega_o^t. \tag{8}$$

Reflection directions can query the environment map for the specular color of static environment light. To handle time-varying lighting in dynamic scenes, we introduce a deformable environment map, detailed in the following section.

### 3.2.2 Deformable Environment Map for Dynamic Lighting.

The concept of a deformable environment map involves treating the vanilla environment map as a canonical environment map and combining it with a deformation field. This approach enables us to model time-varying lighting conditions effectively. To implement this, we first apply positional encoding to the reflection direction $\omega_r^t$ and time $t$. These encoded values are then input into a deformable reflection MLP with parameters $\theta_R$. This process allows us to obtain the deformed reflection residual $\Delta\bar{\omega}_r^t$ for each specified time $t$:

$$\Delta\bar{\omega}_r^t = F_{\theta_R}(\gamma(\omega_r^t), \gamma(t)). \tag{9}$$

Subsequently, we add the deformed reflection residual $\Delta\bar{\omega}_r^t$ to the reflection direction $\omega_r^t$, yielding the deformed reflection direction $\bar{\omega}_r^t$. This can be expressed as:

$$\bar{\omega}_r^t = \Delta\bar{\omega}_r^t + \omega_r^t \tag{10}$$

We can then use this deformed reflection direction $\bar{\omega}_r^t$ to query the canonical environment map, allowing us to obtain time-varying specular color $\mathbf{c}_s^t$. This approach effectively captures the dynamic nature of lighting in the scene while maintaining a stable canonical reference.

### 3.2.3 Color Decomposition and Staged Training Strategy.

We decompose the final color $\mathbf{c}_{\mathbf{final}}^t$ into diffuse and specular components to better distinguish between high and low-frequency information:

$$\mathbf{c}_{\mathbf{final}}^t = \mathbf{c}_d + \mathbf{c}_s^t \odot \mathbf{s}_{\mathbf{tint}}, \tag{11}$$

where $\mathbf{c_d}$ is the diffuse color (using zero-order spherical harmonics as view-independent color), $\mathbf{s_{tint}} \in [0, 1]^3$ is the learnable specular tint stored in each 3D Gaussian, and $\mathbf{c_s}^t$ is the view-dependent color component. To manage the transition from spherical harmonics to $\mathbf{c}_{\mathbf{final}}^t$ and mitigate potential geometry disruptions, in the early specular stage, we fix the deformable Gaussian MLP and most 3D Gaussian attributes, optimizing only zero-order SH, specular tint, and roughness. We temporarily suspend densification during this phase. As $\mathbf{c}_{\mathbf{final}}^t$ becomes more complete, we gradually resume optimization of all parameters and reinstate the densification process.

We further split the specular stage into two parts, applying a coarse-to-fine strategy to the environment map. In the first part, we focus on optimizing the canonical environment map for time-invariant lighting. This establishes a stable foundation for the overall lighting structure. In the second part, we proceed to optimize the deformable reflection MLP for dynamic elements. This approach ensures a more robust learning process, allowing us to capture the static lighting conditions before introducing the complexities of dynamic components.

### 3.3 Physical Normal Estimation

**Challenges in normal estimation for 3D Gaussians.** Normal estimation is crucial for modeling specular objects, as it directly affects surface reflections. However, the discrete nature of 3D Gaussians makes this process challenging, as it typically requires a continuous surface. GaussianShader (Jiang et al., 2023) observed that 3D Gaussians tend to flatten during training, leading to the use of the shortest axis as an initial approximation of the surface normal. To improve accuracy, they introduced a residual normal $\Delta\mathbf{n}$ for each 3D Gaussian to compensate for errors in this approximation. However, this method alone is insufficient for deformed 3D Gaussians, as the residual should vary at each time step. A straightforward approach of rotating the residual based on the quaternion difference between canonical and deformed Gaussians proves inadequate, as it fails to account for shape changes during deformation. If the shortest axis of the canonical 3D Gaussian is no longer the shortest after deformation, this method results in incorrect rotation. Consequently, a more sophisticated approach is needed to accurately model the normals of deformed 3D Gaussians. This approach must consider both the rotation and the change in shape during the deformation process, ensuring accurate normal estimation for dynamic specular objects.

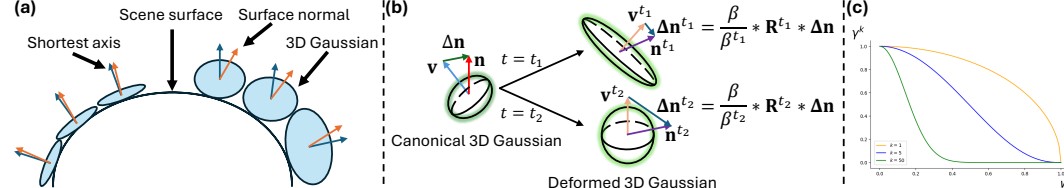

Figure 3: **Normal estimation.** **(a)** shows that flatter 3D Gaussians align better with scene surfaces, their shortest axis closely matching the surface normal. In contrast, less flat 3D Gaussians fit less accurately, with their shortest axis diverging from the surface normal. **(b)** shows that when the deformed 3D Gaussian becomes flatter ($t = t_1$), normal residual $\Delta\mathbf{n}$ is rotated by $\mathbf{R}_1^t$ and scaled down by $\frac{\beta}{\beta_1^t}$, as flatter Gaussians require smaller normal residuals. Conversely, when the deformation results in a less flat shape ($t = t_2$), $\Delta\mathbf{n}$ is rotated by $\mathbf{R}_2^t$ and amplified by $\frac{\beta}{\beta_2^t}$, requiring a larger correction to align the shortest axis with the surface normal. **(c)** shows how $\gamma^k$ changes with $w$ (where $w = \frac{|\mathbf{v}_s^t|}{|\mathbf{v}_l^t|}$) for $k = 1$, $k = 5$, and $k = 50$. Larger $w$ indicates less flat Gaussians, while smaller $w$ represents flatter Gaussians. As $k$ increases, $\gamma^k$ decreases more steeply as $w$ rises. For $k = 5$, we observe a balanced behavior: $\gamma^k$ approaches 1 for low $w$ and 0 for high $w$, providing a nuanced penalty adjustment across different Gaussian shapes.

**Improved rotation calculation for deformed 3D Gaussians.** To overcome the limitations of naive methods and accurately model the normal of deformed 3D Gaussians, we propose using both the shortest and longest axes of canonical and deformed Gaussians to compute the rotation matrix. This approach accounts for both rotation and shape changes during deformation. We first align the deformed Gaussian's axes with those of the canonical Gaussian using the following method:

$$\mathbf{v}_s^t = \begin{cases} \mathbf{v}_s^t & \text{if } \mathbf{v}_s \cdot \mathbf{v}_s^t > 0, \\ -\mathbf{v}_s^t & \text{otherwise.} \end{cases}, \quad \mathbf{v}_l^t = \begin{cases} \mathbf{v}_l^t & \text{if } \mathbf{v}_l \cdot \mathbf{v}_l^t > 0, \\ -\mathbf{v}_l^t & \text{otherwise.} \end{cases}, \tag{12}$$

where $\mathbf{v}_s$ and $\mathbf{v}_l$ represent the shortest and longest axes of canonical 3D Gaussians, while $\mathbf{v}_s^t$ and $\mathbf{v}_l^t$ denote the same for deformed 3D Gaussians. We then construct orthogonal matrices using these aligned axes and their cross products:

$$\mathbf{U} = \begin{bmatrix} \mathbf{v}_s & \mathbf{v}_l & \mathbf{v}_s \times \mathbf{v}_l \end{bmatrix}, \quad \mathbf{V}^t = \begin{bmatrix} \mathbf{v}_s^t & \mathbf{v}_l^t & \mathbf{v}_s^t \times \mathbf{v}_l^t \end{bmatrix}. \tag{13}$$

Finally, we derive the rotation matrix:

$$\mathbf{R}^t = \mathbf{V}^t \mathbf{U}^\top. \tag{14}$$

This method provides a robust solution for calculating the rotation of deformation process, ensuring accurate normal estimation for dynamic specular objects.

**Adjusting normal residuals and ensuring accuracy.** To account for shape changes during deformation, we scale the normal residual based on the ratio of oblateness $\frac{\beta}{\beta^t}$ between canonical and deformed 3D Gaussians.

$$\beta = \frac{|\mathbf{v}_l| - |\mathbf{v}_s|}{|\mathbf{v}_l|}, \quad \beta^t = \frac{|\mathbf{v}_l^t| - |\mathbf{v}_s^t|}{|\mathbf{v}_l^t|}, \tag{15}$$

where $\beta$ and $\beta^t$ represent the oblateness of canonical and deformed 3D Gaussians, respectively. This is because flatter 3D Gaussians tend to align more closely with the surface, meaning their shortest axis becomes more aligned with the surface normal, as shown in Fig. 3(**a**). In such cases, less compensation from the normal residual is needed. Conversely, less flat Gaussians require more compensation, as illustrated in Fig. 3(**b**). We then obtain deformed normal residuals:

$$\Delta\mathbf{n}^t = \frac{\beta}{\beta^t} \mathbf{R}^t \Delta\mathbf{n}. \tag{16}$$

The final normal $\mathbf{n}^t$ is computed by adding this residual to the shortest axis and ensuring outward orientation:

$$\mathbf{n}^t = \Delta\mathbf{n}^t + \mathbf{v}_s^t, \quad \mathbf{n}^t = \begin{cases} \mathbf{n}^t & \text{if } \mathbf{n}^t \cdot \omega_o^t > 0, \\ -\mathbf{n}^t & \text{otherwise.} \end{cases} \tag{17}$$

This approach adjusts for Gaussian flatness and ensures accurate normal estimation.

Table 1: **Quantitative comparison on the NeRF-DS (Yan et al., 2023) dataset.** We report the average PSNR, SSIM, and LPIPS (VGG) of several previous models on test images. The best , the second best , and third best results are denoted by red, orange, yellow.

| | As | | | Basin | | | Bell | | | Cup | | |
|---|---|---|---|---|---|---|---|---|---|---|---|---|
| Method | PSNR↑ | SSIM↑ | LPIPS↓ | PSNR↑ | SSIM↑ | LPIPS↓ | PSNR↑ | SSIM↑ | LPIPS↓ | PSNR↑ | SSIM↑ | LPIPS↓ |
| Deformable 3DGS (Yang et al., 2023c) | 26.04 | 0.8805 | 0.1850 | 19.53 | 0.7855 | 0.1924 | 23.96 | 0.7945 | 0.2767 | 24.49 | 0.8822 | 0.1658 |
| 4DGS (Wu et al., 2023) | 24.85 | 0.8632 | 0.2038 | 19.26 | 0.7670 | 0.2196 | 22.86 | 0.8015 | 0.2061 | 23.82 | 0.8695 | 0.1792 |
| GaussianShader (Jiang et al., 2023) | 21.89 | 0.7739 | 0.3620 | 17.79 | 0.6670 | 0.4187 | 20.69 | 0.8169 | 0.3024 | 20.40 | 0.7437 | 0.3385 |
| GS-IR (Liang et al., 2023d) | 21.58 | 0.8033 | 0.3033 | 18.06 | 0.7248 | 0.3135 | 20.66 | 0.7829 | 0.2603 | 20.34 | 0.8193 | 0.2719 |
| NeRF-DS (Yan et al., 2023) | 25.34 | 0.8803 | 0.2150 | 20.23 | 0.8053 | 0.2508 | 22.57 | 0.7811 | 0.2921 | 24.51 | 0.8802 | 0.1707 |
| HyperNeRF (Park et al., 2021b) | 17.59 | 0.8518 | 0.2390 | 22.58 | 0.8156 | 0.2497 | 19.80 | 0.7650 | 0.2999 | 15.45 | 0.8295 | 0.2302 |
| Ours | 26.80 | 0.8851 | 0.1761 | 19.75 | 0.7922 | 0.1896 | 25.46 | 0.8497 | 0.1600 | 24.65 | 0.8879 | 0.1588 |

| | Plate | | | Press | | | Sieve | | | **Mean** | | |
|---|---|---|---|---|---|---|---|---|---|---|---|---|
| Method | PSNR↑ | SSIM↑ | LPIPS↓ | PSNR↑ | SSIM↑ | LPIPS↓ | PSNR↑ | SSIM↑ | LPIPS↓ | PSNR↑ | SSIM↑ | LPIPS↓ |
| Deformable 3DGS (Yang et al., 2023c) | 19.07 | 0.7352 | 0.3599 | 25.52 | 0.8594 | 0.1964 | 25.37 | 0.8616 | 0.1643 | 23.43 | 0.8284 | 0.2201 |
| 4DGS (Wu et al., 2023) | 18.77 | 0.7709 | 0.2721 | 24.82 | 0.8355 | 0.2255 | 25.16 | 0.8566 | 0.1745 | 22.79 | 0.8235 | 0.2115 |
| GaussianShader (Jiang et al., 2023) | 14.55 | 0.6423 | 0.4955 | 19.97 | 0.7244 | 0.4507 | 22.58 | 0.7862 | 0.3057 | 19.70 | 0.7363 | 0.3819 |
| GS-IR (Liang et al., 2023d) | 15.98 | 0.6969 | 0.4200 | 22.28 | 0.8088 | 0.3067 | 22.84 | 0.8212 | 0.2236 | 20.25 | 0.7796 | 0.2999 |
| NeRF-DS (Yan et al., 2023) | 19.70 | 0.7813 | 0.2974 | 25.35 | 0.8703 | 0.2552 | 24.99 | 0.8705 | 0.2001 | 23.24 | 0.8384 | 0.2402 |
| HyperNeRF (Park et al., 2021b) | 21.22 | 0.7829 | 0.3166 | 16.54 | 0.8200 | 0.2810 | 19.92 | 0.8521 | 0.2142 | 19.01 | 0.8167 | 0.2615 |
| Ours | 20.84 | 0.8180 | 0.2198 | 26.49 | 0.8665 | 0.1889 | 25.22 | 0.8712 | 0.1513 | 24.17 | 0.8529 | 0.1778 |

Figure 4: **Qualitative comparison on the NeRF-DS Yan et al. (2023) dataset.**

### 3.4 Loss Functions

**Normal regularization.** To allow the normal residual to correct the normal while not excessively influencing the optimization of the shortest axis towards the surface normal, we introduce a penalty term for the normal residual:

$$\mathcal{L}_{\text{reg}} = \gamma^k \|\Delta\mathbf{n}\|_2^2 \quad \text{where} \quad \gamma = \sqrt{1 - \frac{|\mathbf{v}_s^t|^2}{|\mathbf{v}_l^t|^2}}. \tag{18}$$

In our experiments, we set $k = 5$. When $k = 5$, less flatter 3D Gaussians have $\gamma^k$ approaching 0. Their shortest axis aligns poorly with the surface normal, requiring more normal residual correction and smaller penalties. Conversely, flatter Gaussians have $\gamma^k$ near 1. Their shortest axis aligns better with the surface normal, needing less normal residual correction and allowing larger penalties, as shown in Fig. 3(**c**).

**Total training loss.** To refine all parameters in the dynamic and specular stages, we employ the total training loss:

$$\mathcal{L} = \mathcal{L}_{\text{color}} + \lambda_{\text{normal}}\mathcal{L}_{\text{normal}} + \mathcal{L}_{\text{reg}}, \tag{19}$$

where $\mathcal{L}_{\text{color}}$ and $\mathcal{L}_{\text{normal}}$ are obtained as described in Section 3.2.1. In our experiments, we set $\lambda_{\text{normal}} = 0.01$.

## 4 Experiments

### 4.1 Evaluation Results

We evaluate our method on two real-world datasets: NeRF-DS dataset (Yan et al., 2023) and HyperNeRF dataset (Park et al., 2021b).

Table 2: **Quantitative comparison on the NeRF-DS (Yan et al., 2023) dataset with our labeled dynamic specular masks.** We report PSNR, SSIM, and LPIPS (VGG) of previous methods on dynamic specular objects using the dynamic specular objects mask generated by Track Anything (Yang et al., 2023a). The  best , the  second best , and  third best  results are denoted by red, orange, yellow.

| | As | | | Basin | | | Bell | | | Cup | | |
|---|---|---|---|---|---|---|---|---|---|---|---|---|
| Method | PSNR↑ | SSIM↑ | LPIPS↓ | PSNR↑ | SSIM↑ | LPIPS↓ | PSNR↑ | SSIM↑ | LPIPS↓ | PSNR↑ | SSIM↑ | LPIPS↓ |
| Deformable 3DGS (Yang et al., 2023c) | 24.14 | 0.7432 | 0.2957 | 17.45 | 0.5530 | 0.3138 | 19.42 | 0.5516 | 0.2940 | 20.10 | 0.5446 | 0.3312 |
| 4DGS (Wu et al., 2023) | 22.70 | 0.6993 | 0.3517 | 16.61 | 0.4797 | 0.4084 | 14.64 | 0.2596 | 0.4467 | 18.90 | 0.4132 | 0.4032 |
| GaussianShader (Jiang et al., 2023) | 19.27 | 0.5652 | 0.5232 | 15.71 | 0.4163 | 0.5941 | 12.10 | 0.1676 | 0.6764 | 14.90 | 0.3634 | 0.6146 |
| GS-IR (Liang et al., 2023d) | 19.32 | 0.5857 | 0.4782 | 15.21 | 0.4009 | 0.5644 | 12.09 | 0.1757 | 0.6722 | 14.80 | 0.3445 | 0.6046 |
| NeRF-DS (Yan et al., 2023) | 23.67 | 0.7478 | 0.3635 | 17.98 | 0.5537 | 0.4211 | 14.73 | 0.2439 | 0.5931 | 19.95 | 0.5079 | 0.3494 |
| HyperNeRF (Park et al., 2021b) | 17.37 | 0.6934 | 0.3834 | 18.75 | 0.5671 | 0.4125 | 13.93 | 0.2292 | 0.6051 | 15.07 | 0.4860 | 0.4183 |
| Ours | 24.51 | 0.7534 | 0.2896 | 17.71 | 0.5675 | 0.3048 | 19.60 | 0.5680 | 0.2862 | 20.28 | 0.5473 | 0.3176 |

| | Plate | | | Press | | | Sieve | | | Mean | | |
|---|---|---|---|---|---|---|---|---|---|---|---|---|
| Method | PSNR↑ | SSIM↑ | LPIPS↓ | PSNR↑ | SSIM↑ | LPIPS↓ | PSNR↑ | SSIM↑ | LPIPS↓ | PSNR↑ | SSIM↑ | LPIPS↓ |
| Deformable 3DGS (Yang et al., 2023c) | 16.12 | 0.5192 | 0.3544 | 19.64 | 0.6384 | 0.3268 | 20.74 | 0.5283 | 0.3109 | 19.66 | 0.5826 | 0.3181 |
| 4DGS (Wu et al., 2023) | 13.93 | 0.4095 | 0.4229 | 20.17 | 0.5434 | 0.4339 | 19.70 | 0.4498 | 0.3879 | 18.09 | 0.4649 | 0.4078 |
| GaussianShader (Jiang et al., 2023) | 9.87 | 0.2992 | 0.6812 | 16.84 | 0.4408 | 0.6093 | 16.19 | 0.3241 | 0.5862 | 14.98 | 0.3681 | 0.6121 |
| GS-IR (Liang et al., 2023d) | 11.09 | 0.3254 | 0.6270 | 16.43 | 0.4083 | 0.5776 | 16.42 | 0.3339 | 0.5749 | 15.05 | 0.3678 | 0.5856 |
| NeRF-DS (Yan et al., 2023) | 14.80 | 0.4518 | 0.3987 | 19.77 | 0.5835 | 0.5035 | 20.28 | 0.5173 | 0.4067 | 18.74 | 0.5151 | 0.4337 |
| HyperNeRF (Park et al., 2021b) | 16.03 | 0.4629 | 0.3775 | 14.10 | 0.5365 | 0.5023 | 18.39 | 0.5296 | 0.3949 | 16.23 | 0.5007 | 0.4420 |
| Ours | 16.53 | 0.5369 | 0.3041 | 21.70 | 0.6630 | 0.3252 | 20.36 | 0.5089 | 0.3190 | 20.10 | 0.5921 | 0.3066 |

Figure 5: **Qualitative comparison on NeRF-DS dataset with labeled dynamic specular masks.**

**Entire scene of the NeRF-DS dataset.** The NeRF-DS dataset (Yan et al., 2023) is a monocular video dataset comprising seven real-world scenes from daily life, featuring various types of moving or deforming specular objects. We compare our method with the most relevant state-of-the-art approaches. As shown in Tab. 1 and Fig. 4, the quantitative results demonstrate that our method decisively outperforms baselines in reconstructing and rendering real-world highly reflective dynamic specular scenes.

**Dynamic specular object of NeRF-DS dataset.** Since each scene in the NeRF-DS dataset (Yan et al., 2023) contains not only dynamic specular objects but also static background objects, we use Track Anything (Yang et al., 2023a) to obtain masks for the dynamic specular objects. This allows us to evaluate only the dynamic specular objects. As shown in Tab. 2 and Fig. 5, our method outperforms baselines when evaluating the dynamic specular objects in these monocular sequences.

**HyperNeRF dataset.** The HyperNeRF dataset, while also containing real-world dynamic scenes, does not include specular objects. As shown in Tab. 3 and appendix Fig. 14, the results demonstrate that we are on par with state-of-the-art techniques for rendering novel views and our method's performance is not limited to shiny scenes.

This strong performance across different types of real-world datasets further confirms the effectiveness of our approach in handling a wide range of scene characteristics. The success can be attributed to our key innovations: physical normal estimation, deformable environment map, and coarse-to-fine training strategy, which together enable robust reconstruction and rendering of diverse scenes. Notably, unlike NeRF-DS, our approach does not require mask supervision to clearly distinguish between static and dynamic objects, as illustrated in Fig. 6. Additionally, Fig. 7 illustrates our method's decomposition results. As shown, our approach consistently achieves a realistic separation of specular and diffuse components across different scenes in the NeRF-DS dataset.

Table 3: **Quantitative comparison on the HyperNeRF (Park et al., 2021b) dataset.** We report the average PSNR, SSIM, and LPIPS (VGG) of several previous models. The best , the second best , and third best results are denoted by red, orange, yellow.

| | Broom | | | 3D printer | | | Chicken | | | Peel Banana | | | **Mean** | | |
|---|---|---|---|---|---|---|---|---|---|---|---|---|---|---|---|
| Method | PSNR↑ | SSIM↑ | LPIPS↓ | PSNR↑ | SSIM↑ | LPIPS↓ | PSNR↑ | SSIM↑ | LPIPS↓ | PSNR↑ | SSIM↑ | LPIPS↓ | PSNR↑ | SSIM↑ | LPIPS↓ |
| Deformable 3DGS (Yang et al., 2023c) | 22.35 | 0.4952 | 0.5148 | 21.47 | 0.6921 | 0.2147 | 23.55 | 0.6747 | 0.2334 | 21.28 | 0.5302 | 0.4472 | 22.16 | 0.5981 | 0.3525 |
| 4DGS (Wu et al., 2023) | 21.21 | 0.3555 | 0.5669 | 21.90 | 0.6993 | 0.3198 | 28.69 | 0.8143 | 0.2772 | 27.77 | 0.8431 | 0.2049 | 24.89 | 0.6781 | 0.3422 |
| GaussianShader (Jiang et al., 2023) | 17.21 | 0.2263 | 0.5812 | 17.31 | 0.5926 | 0.5054 | 19.70 | 0.6520 | 0.5004 | 19.99 | 0.7097 | 0.3308 | 18.55 | 0.5452 | 0.4795 |
| GS-IR (Liang et al., 2023d) | 20.46 | 0.3420 | 0.5229 | 18.24 | 0.5745 | 0.5204 | 20.64 | 0.6592 | 0.4536 | 20.15 | 0.7159 | 0.3021 | 19.87 | 0.5729 | 0.4498 |
| NeRF-DS (Yan et al., 2023) | 22.37 | 0.4371 | 0.5694 | 22.16 | 0.6973 | 0.3134 | 27.32 | 0.7949 | 0.3139 | 22.75 | 0.6328 | 0.3919 | 23.65 | 0.6405 | 0.3972 |
| HyperNeRF (Park et al., 2021b) | 20.72 | 0.4276 | 0.5773 | 21.94 | 0.7003 | 0.3090 | 27.40 | 0.8013 | 0.3052 | 22.36 | 0.6257 | 0.3956 | 23.11 | 0.6387 | 0.3968 |
| Ours | 22.04 | 0.5145 | 0.4494 | 19.96 | 0.6444 | 0.2397 | 22.20 | 0.6203 | 0.1970 | 27.34 | 0.8895 | 0.1290 | 22.89 | 0.6672 | 0.2538 |

Table 4: **Ablation studies on different coarse to fine training strategy stages.**

| Stage | PSNR ↑ | SSIM ↑ | LPIPS ↓ |
|---|---|---|---|
| Static | 20.26 | 0.7785 | 0.2953 |
| St. + Dynamic | 24.02 | 0.8508 | 0.1831 |
| St. + Dy. + Specular | **24.17** | **0.8529** | **0.1778** |

Table 5: **Ablation study on coarse-to-fine and losses.**

| C2F | $\mathcal{L}_{normal}$ | $\mathcal{L}_{reg}$ | $\gamma^k$ | PSNR↑ | SSIM↑ | LPIPS↓ |
|---|---|---|---|---|---|---|
| | ✓ | ✓ | ✓ | 23.16 | 0.8294 | 0.2156 |
| ✓ | | | | 23.40 | 0.8277 | 0.2278 |
| ✓ | ✓ | | | 24.15 | 0.8510 | 0.1845 |
| ✓ | ✓ | ✓ | | 24.09 | 0.8515 | 0.1818 |
| ✓ | ✓ | ✓ | ✓ | **24.17** | **0.8529** | **0.1778** |

Table 6: **Ablation studies on SH, Static and Deformable environment map.**

| | PSNR ↑ | SSIM ↑ | LPIPS ↓ |
|---|---|---|---|
| SH | 23.63 | 0.8453 | 0.1844 |
| Static Env. map | 24.02 | 0.8508 | 0.1831 |
| Deformable Env. map | 24.17 | 0.8529 | 0.1778 |

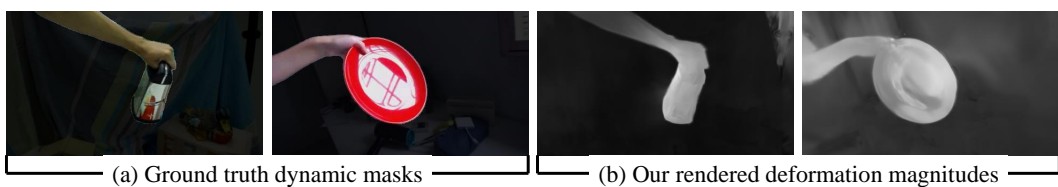

(a) Ground truth dynamic masks — (b) Our rendered deformation magnitudes

Figure 6: **Visualization our deformation magnitudes.** (a) The left side shows the ground truth of the dynamic object, while (b) on the right side, we render the magnitude of the output of the position residual by our deformable Gaussian MLP. The brighter areas indicate greater movement of the 3D Gaussians. The figure shows that even without mask supervision, our method can still effectively distinguish which objects are dynamic.

## 4.2 ABLATION STUDY

**Different coarse to fine training strategy stages.** As shown in the Tab. 4 and Fig. 8 , each stage contributes effectively to the model's performance. The Dynamic stage enhances object stability compared to the Static stage alone, while the Specular stage improves reflection clarity beyond the combined Static and Dynamic stages. This coarse-to-fine approach establishes a stable geometric foundation before addressing complex specular effects. Note that the total iterations for each row in the Tab. 4 are 40,000.

**Ablation study on coarse-to-fine, and loss function.** The model's performance was evaluated without key components: the coarse-to-fine training strategy, normal loss $\mathcal{L}_{normal}$, normal regularization $\mathcal{L}_{reg}$, and $\gamma^k$. Fig. 9 and Tab. 5 illustrate the effects of these omissions. Without the coarse-to-fine approach, the model, trained directly at the specular stage, produces incomplete scene geometry, affecting environment map queries for specular color. Omitting normal loss $\mathcal{L}_{normal}$ removes direct supervision on normals, impeding geometric refinement and reducing rendering quality. This also leads to inaccurate reflection directions and less precise specular colors. Removing normal regularization $\mathcal{L}_{reg}$ allows the normal residual to dominate normal optimization, resulting in insufficient optimization of the 3D Gaussians' shortest axis towards the correct normal , which in turn reduces the rendering quality. Without $\gamma^k$ in normal regularization, the normal residual decreases for both non-flattened and flat Gaussians. This particularly affects less flat 3D Gaussians whose shortest axis significantly deviates from the surface normal. The insufficient normal residual correction causes these 3D Gaussians' shortest axes to deviate greatly from their original direction in an attempt to align with the surface normal, ultimately reducing rendering quality.

**Ablation study on SH, Static environment map, and Deformable enviorment map.** Fig. 10 and Tab. 6 demonstrate the superiority of the deformable environment map over the static environment

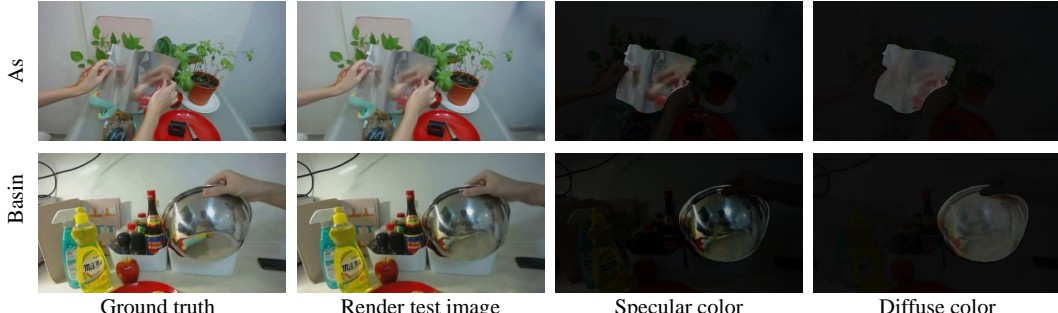

Figure 7: **Visualization our specular and diffuse color.** Specular regions are emphasized while non-specular areas are dimmed to highlight the results of specular region color decomposition.

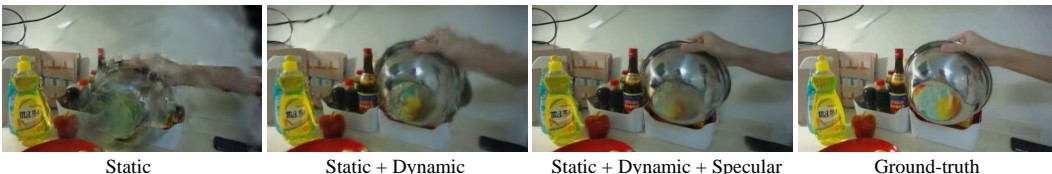

Figure 8: **Qualitative comparison of each training stage in our coarse-to-fine approach.**

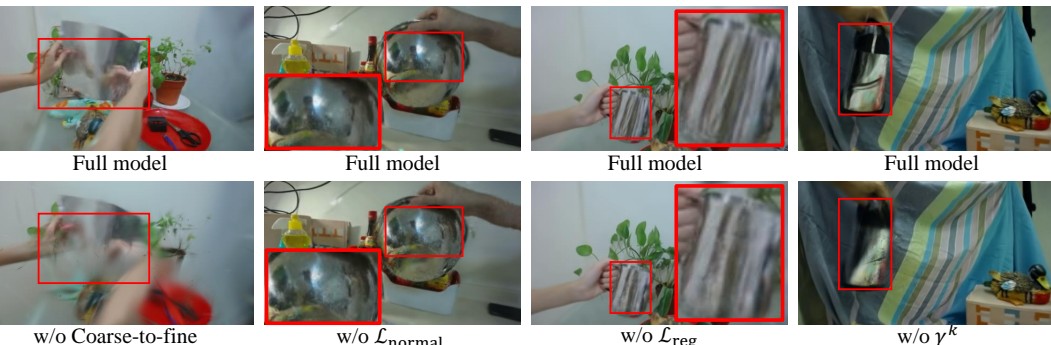

Figure 9: **Qualitative comparison of ablation study without different components.**

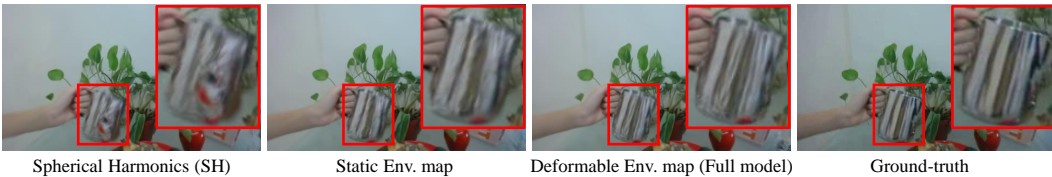

Figure 10: **Qualitative comparison of ablation study on SH, Static environment map, and Deformable enviorment map.**

map, which in turn outperforms Spherical Harmonics (SH). SH struggles to accurately model high-frequency specular colors. While the static environment map can model high-frequency colors, it is best suited for static lighting conditions. In contrast, the deformable environment map excels in modeling time-varying lighting, offering superior performance for dynamic scenes.

## 5 CONCLUSION

SpectroMotion enhances 3D Gaussian Splatting for dynamic specular scenes by combining specular rendering with deformation fields. Using normal residual correction, coarse-to-fine training, and deformable environment map, it achieves superior accuracy and visual quality in novel view synthesis, outperforming existing methods while maintaining geometric consistency.

**Limitations.** Though we stabilize the entire scene's geometry using a coarse-to-fine training strategy, some failure cases still occur. Please refer to the appendix for visual results of failure cases.

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

# A APPENDIX / SUPPLEMENTAL MATERIAL

## A.1 IMPLEMENTATION DETAILS

We use PyTorch as our framework and 3DGS (Kerbl et al., 2023) as our codebase. Our coarse-to-fine approach is divided into three sequential stages: static, dynamic, and specular. In the static stage, we train the vanilla 3D Gaussian Splatting (3DGS) for 3000 iterations to stabilize the static geometry. The dynamic stage then introduces a deformable Gaussian MLP to model dynamic objects. We first optimize both the canonical Gaussians and the deformable Gaussian MLP for 3000 iterations until the scene becomes relatively stable. Subsequently, we introduce $\mathcal{L}_{\text{normal}}$, enabling simultaneous

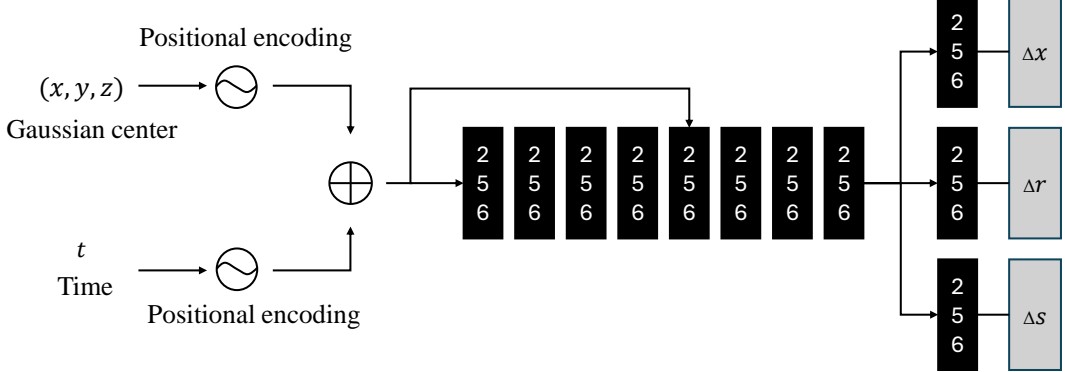

Figure 11: **Architecture of the deformable Gaussian MLP**

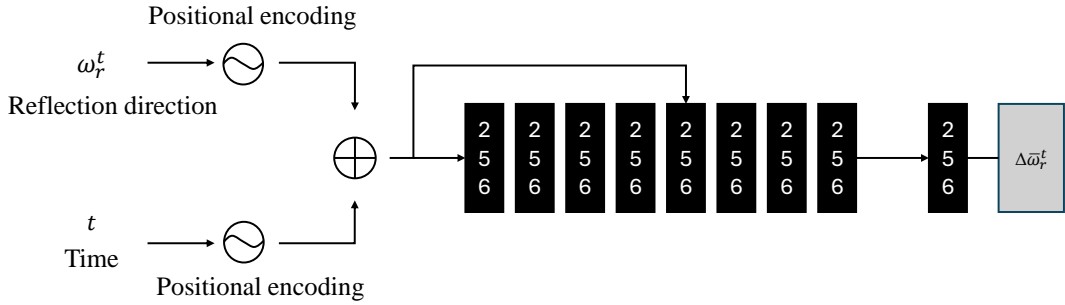

Figure 12: **Architecture of the deformable reflection MLP**

optimization of the scene's normal and depth, further refining the geometry for another 3000 iterations. After the dynamic stage concludes, we transition to the specular stage, which involves changing the color representation from complete spherical harmonics to $\mathbf{c_{final}}$. To mitigate potential geometry disruptions due to the initially incomplete $\mathbf{c_{final}}$, we fix the deformable Gaussian MLP and all 3D Gaussian attributes except for zero-order SH, specular tint, and roughness, while temporarily suspending densification. After 6000 iterations, once $\mathbf{c_{final}}$ becomes more complete, we resume optimization of all parameters and reinstate the densification process. Concurrently, during the first 2000 iterations of the specular stage, we optimize only the canonical environment map to learn time-invariant lighting. For the canonical environment, we use $6 \times 128 \times 128$ learnable parameters. Subsequently, we begin optimizing the deformable reflection MLP to capture time-varying lighting effects. We train total iterations 40,000. All training and rendering are conducted on an NVIDIA RTX 4090 GPU.

### A.1.1 NETWORK ARCHITECTURE OF THE DEFORMABLE GAUSSIAN MLP AND DEFORMABLE REFLECTION MLP

We use deformable Gaussian MLP to predict each coordinate of 3D Gaussians and time to their corresponding deviations in position, rotation, and scaling. As shown in Fig. 11, the MLP initially processes the input through eight fully connected layers that employ ReLU activations, featuring 256-dimensional hidden layers, and outputs a 256-dimensional feature vector. This vector is then passed through three additional fully connected layers combined with ReLU activation to separately output the offsets over time for position, rotation, and scaling. Notably, similar to NeRF, the feature vector and the input are concatenated in the fourth layer. For the deformable reflection MLP, we utilize the same network architecture, as shown in Fig. 12.

## A.2 Additional Experiment Results

We provide an HTML interface in the supplementary material zip file for browser-rendered video results of all compared methods. This includes qualitative comparisons on the NeRF-DS dataset for each scene, as shown in Fig. 13, as well as qualitative comparisons on the NeRF-DS dataset for each scene with labeled dynamic specular masks, as shown in Fig. 15. Additionally, failure cases are presented in Fig. 16.

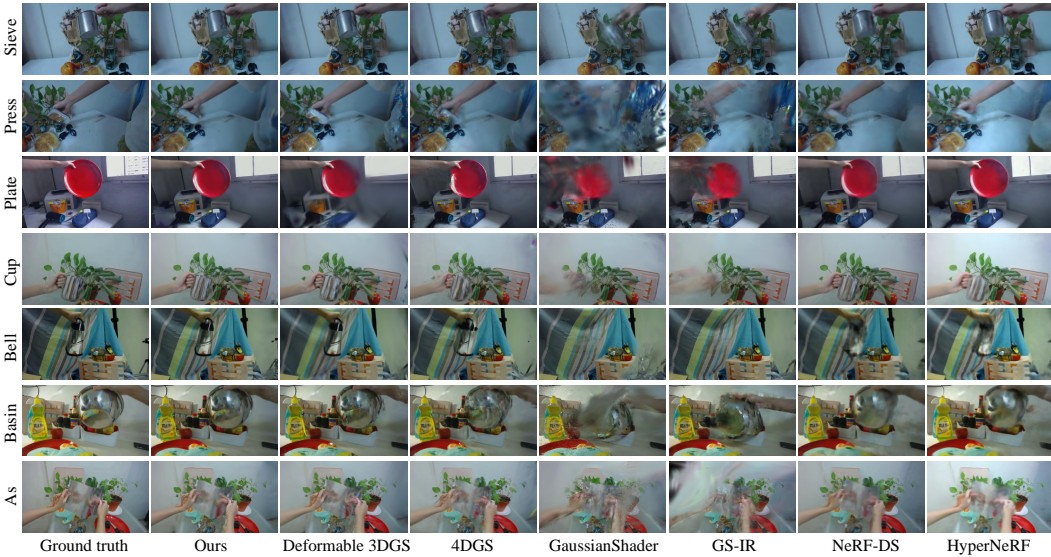

Figure 13: **Qualitative comparison on NeRF-DS dataset per-scene.**

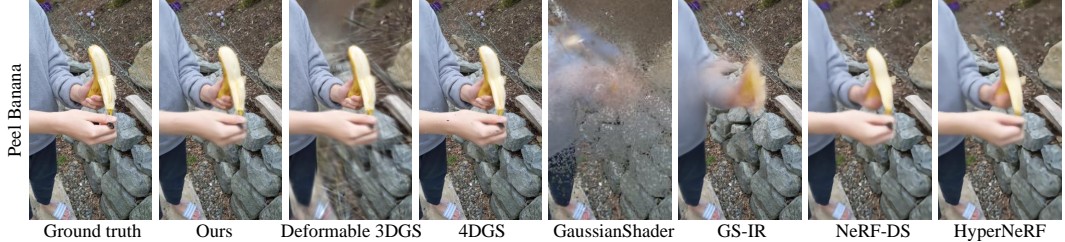

Figure 14: **Qualitative comparison on the HyperNeRF Park et al. (2021b) dataset.**

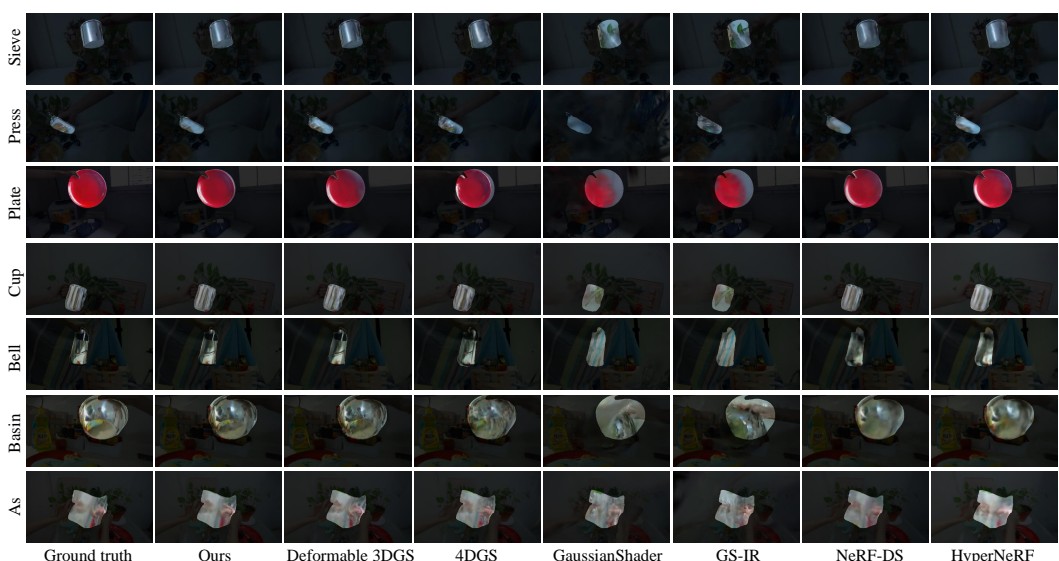

Figure 15: **Qualitative comparison on NeRF-DS dataset per-scene with labeled dynamic specular masks.**

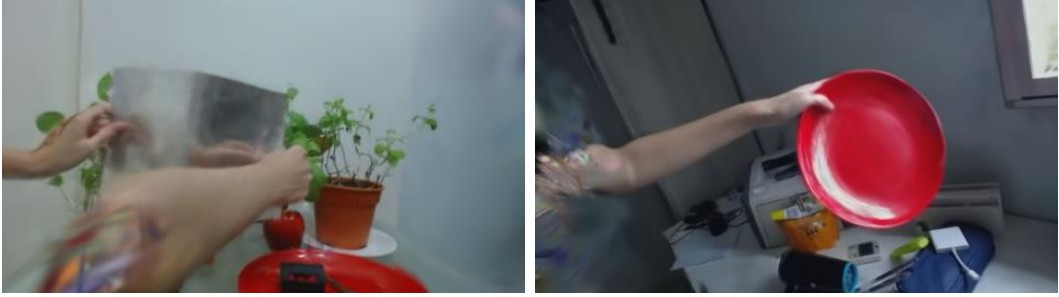

Figure 16: **Failure cases.** In some dramatic scenes, relying solely on the Deformable Gaussian MLP is insufficient, such as when an arm enters or exits the scene, leading to many floaters occurring.

