# OpenReview forum: "SpectroMotion: Dynamic 3D Reconstruction of Specular Scenes"
_ICLR.cc/2025/Conference — ICLR 2025 Conference Withdrawn Submission_

### Official Review · Reviewer_VUVz · 2024-10-19

**Soundness:** 4
**Presentation:** 2
**Contribution:** 2
**Rating:** 5
**Confidence:** 5

**Summary:**

This paper aims to address the modeling of dynamic scenes with specular highlights and reflections. Building on Deformable-GS, the authors introduce the Cook-Torrance BRDF to compute the high-frequency specular components. Additionally, they incorporate techniques such as robust rotation calculation and normal residuals to improve geometric accuracy, which in turn enhances the accuracy of reflective view directions and overall model fitting capability. The experimental results show that SpectroMotion achieves impressive results on NeRF-DS dataset. The robust results on HyperNeRF also demonstrate that the approach can be applied to more general scenarios.

**Strengths:**

- This paper is well-written and easy to understand.
- Using zero-order spherical harmonics (SH) and Cook-Torrance BRDF to model diffuse and specular respectively makes sense.
- Using normal residuals to flatten the Gaussian, thereby improving the accuracy of normal estimation in dynamic scenes, is insightful for the community.
- The visual results have improved on the NeRF-DS and HyperNeRF datasets.

**Weaknesses:**

1. Inappropriate reference:
   - In Line 103, strictly speaking, Gaussian-DR is not a PBR method, because its approach does not incorporate the rendering equation.
   - Missing reference: Relightable 3D Gaussians: Realistic Point Cloud Relighting with BRDF Decomposition and Ray Tracing, by Gao et al., ECCV 2024.
2. This method, in principle, lacks novelty and can be seen as a combination of Deformable-GS and Gaussian-Shader to some extent.
3. The introduction of BRDF calculations and the deformation reflection MLP will further increase the rendering burden, making real-time rendering difficult to achieve.
4. The quantitative comparisons on the HyperNeRF dataset are not convincing. Inaccuracies in HyperNeRF's camera poses lead to slight misalignments between rendered images and the ground truth, as noted in the original HyperNeRF paper. This results in abnormal outcomes for metrics sensitive to minor shifts—such as PSNR, SSIM, and LPIPS—where sharp but misaligned images score lower than blurry ones. In 3DGS, these inaccurate poses further cause an excessive increase in the number of Gaussians, leading to out-of-memory issues. Therefore, the quantitative metrics for HyperNeRF are not very meaningful, and greater emphasis should be placed on visual quality.

**Questions:**

1. In Table 1, why are the metrics so low for all methods except SpectroMotion? This is inconsistent with the results of Deformable-GS, especially for HyperNeRF and NeRF-DS. Did the authors use different resolutions or evaluation metrics?
2. In the paper, I couldn’t find a clear explanation of how $c_s$ is calculated. It’s unclear whether the authors followed the approach from GaussianShader, where $c_s$ is derived using Eq. 7, or if they adopted the Gaussian-DR method, where the specular color is directly obtained through querying the environment map.

---

### Official Review · Reviewer_71AC · 2024-10-30

**Soundness:** 2
**Presentation:** 3
**Contribution:** 2
**Rating:** 5
**Confidence:** 4

**Summary:**

This paper proposes an extension to the existing deformable 3DGS methods to improve the performance for dynamic scenes with moving specular objects. It identifies the limited capabilities of existing dynamic 3DGS on specular objects and introduces a 3-stage training pipeline. First it trains the static part of the scene, and then adds in the dynamic part using deformation MLP. Lastly, it adds in the specular color estimation using the reflected ray direction to query a deformable environment map. To ensure the quality of surface normal estimation, it uses a consistency loss between estimated gs normal with the depth gradient. To ensure surface normal correctness under deformation, it introduces a normal correction term based on the flatness of the Gaussian. To model changing environment lighting, a deformation field for reflection direction is introduced to change the reflected direction when querying the environment map. As a result, the performance on scenes with move specular objects demonstrates a minor quality improvement.

**Strengths:**

**Motivation**
* This paper correctly identifies the challenging moving specular scene reconstruction using the existing methods. It extends the NeRF-DS problem settings using the latest deformable 3DGS methods to improve the performance.

**Method**
* The proposed method utilizes the PBR to improve the modeling of specular colors under motion. This is very challenging because of the difficulty of accurately reconstructing the surface's normal direction and environment lighting under a monocular dynamic setting.
* The paper uses surface normal self-consistency and proposes surface normal residual correction algorithms to improve the surface normal prediction.
* To tackle the environment lighting estimation, this paper proposes a novel dynamic environment mapping method to extend the idea of the deformation field to the reflected direction when querying the environment map.

**Experiment**
* The paper demonstrates a minor improvement over the existing dynamic 3DGS methods and NeRF-based methods designed for dynamic specular objects. Both qualitative and quantitative results support this improvement to a certain extent.

**Weaknesses:**

**Method**
* My main concern regarding the proposed method is the theoretical validity of the deformable environment map.
* First, an important assumption of using an environment map to model a specular surface is that the environment should be far away from the surface. However, the reflective surface present in the datasets is mainly reflecting objects close to it. This assumption is further violated when this specular surface starts moving. This is all because the query of environment mapping considers only direction but not the position of the surface point. So I am not very sure whether the use of an environment map for such scenes is a good choice.
* Second, the deformable environment map lacks theoretical support. The paper justifies it as a model for the changing environment lighting, but the scenes used in the paper are of static environment lighting. Although the ablation study shows a minor improvement in the performance when using this deformable environment map, but it could be because of the following three reasons: 1. compensating the inaccurate surface normal estimated, 2. compensating the relative motion of the object with its surrounding due to the incorrect environment mapping assumption mentioned above, 3. MLP carries more modeling capacity to fit the specular lighting.
* Third, the paper claims the use of the Cook-Torrance reflectance model for PBR, but this model requires an expensive integration in the hemisphere based on the BRDF. This raises a few issue: 1. the paper seems to miss out how this BRDF is estimated, 2. the paper seems to miss out how this integration is done, 3. the paper did not report the inference time which could be a concern due to this expensive integration.

**Experiments**
* The performance improvement on the NeRF-DS dataset is not very significant.
* The performance improvement on the NeRF-DS dataset specular region is less significant than the overall improvement, which indicates that the performance improvement on the non-specular regions is contributing more to the overall performance.
* The PSNR for HyperNeRF dataset is lower than the 4DGS by 2dB, but the LPIPS is better. This is not impossible, but some explanation would be helpful.

I believe this paper is a useful extension to the dynamic specular object reconstruction, but the methods introduced are not very well justified with theoretical reasoning. The small performance improvement also decreases the contribution of this paper.

**Questions:**

1. Please justify the use of deformable environment map based on the doubts I listed in the weakness section.
2. Please explain the PBR process in more detail, specifcally how the integration is done, how the BRDF is estimated, and how this affects the inference time.
3. Please explain the weird performance difference on HyperNeRF dataset.

---

### Official Review · Reviewer_yNiG · 2024-11-02

**Soundness:** 3
**Presentation:** 3
**Contribution:** 2
**Rating:** 5
**Confidence:** 4

**Summary:**

The paper proposes SpectroMotion, a 3-stage pipeline to reconstruct specular dynamic scenes using 3D Gaussian Splatting representation and physically-based rendering. The paper leverages deformable environment map to model the time-varying lighting, and proposes a more physically-correct normal estimation for deformed Gaussians using rotation and residual adjustments. Experimental results show that the proposed method outperforms state-of-the-art baselines.

**Strengths:**

1. The paper is generally well-written, easy to follow and understand.
2. The proposed deformable environment map and physical normal estimation is reasonable and meaningful.
3. The experimental results show improvements over baselines.

**Weaknesses:**

1. Complex pipeline. The proposed pipeline consists of 3 stages, which may limit its performance and scalability.
2. The proposed physical normal estimation from 3D Gaussians is good, but recent methods (e.g. 2D Gaussian Splatting) have proposed a more general and intuitive solution for well-defined Gaussian normals. With these more simple and effective representations, I doubt the necessity of designing complex corrections for the ill-defined 3D Gaussian normal estimation problem.
3. Missing ablation study. The physical normal estimation is a key design of the method, but the paper doesn't present an ablation study on the effectiveness of this design.
4. Limited novelty. The major parts of the pipeline, including the 3DGS representation, the deformable MLP, and the physically-based rendering model, are not new in this research field. Despite some task-specific designs, I'm not sure if the novelty and innovation can reach the acceptance level of ICLR.

**Questions:**

1. The static stage. Does the pipeline directly optimize a static 3DGS from the input images, even the scene behind the images is dynamic? This will cause significant artifacts in the dynamic parts (in Fig. 8), so why not directly start from the dynamic stage?
2. In Fig. 7, seems like the decomposition between diffuse color and specular color does not work very well.
3. GaussianShader and GS-IR are designed for static scenes, not for dynamic scenes. Is it fair to compare with them?

---

### Official Review · Reviewer_o9QB · 2024-11-04

**Soundness:** 2
**Presentation:** 2
**Contribution:** 2
**Rating:** 5
**Confidence:** 3

**Summary:**

This paper presents an extension of Gaussian Splatting to effectively model scenes containing dynamic and specular objects. Building upon the classic 3DGS pipeline, the authors introduce separete MLPs to model the deformation fields of Gaussians and environment maps. A novel technique for normal estimation is also proposed, which ensures improved accuracy in normal estimation, subsequently enhancing the computation of specular colors. The approach involves joint optimization of Gaussian parameters, MLP parameters, and a trainable environment map, organized into distinct training stages to promote efficient convergence. Experimental results demonstrate superior performance on popular datasets across different metrics.

**Strengths:**

This work introduces an interesting solution for modeling both dynamic and specular objects, incorporating an improved normal estimation technique alongside normal regularization, which are particularly noteworthy contributions. According to the reported results, this method demonstrates enhanced performance in novel view synthesis across multiple datasets.

**Weaknesses:**

1. While the method claims to model both specular and dynamic objects, the results presented from NeRF-DS appear relatively static to me, despite the significant camera motion in these data. I would be particularly interested in seeing results that exhibit greater dynamics, similar to those from D-NeRF Synthetic (e.g., Lego and Jumpingjacks) or HyperNeRF (e.g., cut-lemon and expresso).
2. The paper introduces a deformable environment map to account for dynamic lighting; however, the visualizations provided still seem quite static. Although the ablation study demonstrates the effectiveness of this deformable environment map, I would appreciate additional visualizations where the dynamic lighting effect is more obvious, such as scenes where the light source is moving. Additionally, based on my understanding, this formulation does not seem to handle changes in light color, such as a transition from red to blue light. Please correct me if I am mistaken.
3. To validate the accuracy of normal estimation, it would be beneficial to see more examples of rendered normal maps from various scenes. Currently, these are only available in Figures 1 and 2. Some synthetic data (e.g., the Shiny Blender dataset) could also be used to quantatively evaluate the normal estimation accuracy.
4. Given the explicit decomposition of lighting and material in the proposed method, it has potential applications in areas like material editing and relighting. I believe including examples of these applications would further strengthen the claims made in the paper, particularly by illustrating the variations in specular patterns under different lighting conditions. For example, could the roughness of objects be adjusted to make them appear more specular? Additionally, using different environment maps (e.g., indoor and outdoor HDRI maps) could illustrate variations in rendering.

**Questions:**

Given my concerns listed in Weaknesses, I'm currently leaning towards the negative side about this submission. Besides, I believe the writeup could be further polished to make it easier to follow, like changing the order of certain sections. For example, $n^t$ is introduced in Section 3.2.1, but its formal computation is not explained until Section 3.3. Moving the explanation of $n^t$ computation earlier could facilitate better comprehension. I look forward to the authors' rebuttal to see if these concerns can be adequately addressed.

---

### Note · Authors · 2024-11-14

I have read and agree with the venue's withdrawal policy on behalf of myself and my co-authors.